# The Molecular Subtype of Adult Acute Lymphoblastic Leukemia Samples Determines the Engraftment Site and Proliferation Kinetics in Patient-Derived Xenograft Models

**DOI:** 10.3390/cells11010150

**Published:** 2022-01-03

**Authors:** Anna Richter, Catrin Roolf, Anett Sekora, Gudrun Knuebel, Saskia Krohn, Sandra Lange, Vivien Krebs, Bjoern Schneider, Johannes Lakner, Christoph Wittke, Christoph Kiefel, Irmela Jeremias, Hugo Murua Escobar, Brigitte Vollmar, Christian Junghanss

**Affiliations:** 1Department of Medicine, Clinic III, Hematology, Oncology, Palliative Medicine, Rostock University Medical Center, Ernst-Heydemann-Str. 6, 18057 Rostock, Germany; catrin.roolf@gmail.com (C.R.); anett.sekora@med.uni-rostock.de (A.S.); Gudrun.knuebel@med.uni-rostock.de (G.K.); saskia.krohn@med.uni-rostock.de (S.K.); Sandra.lange@med.uni-rostock.de (S.L.); vivien.krebs@med.uni-rostock.de (V.K.); johannes.lakner@med.uni-rostock.de (J.L.); christoph.wittke@med.uni-rostock.de (C.W.); christoph.kiefel@med.uni-rostock.de (C.K.); hugo.murua.escobar@med.uni-rostock.de (H.M.E.); christian.junghanss@med.uni-rostock.de (C.J.); 2Institute of Pathology, Rostock University Medical Center, Strempelstr. 14, 18057 Rostock, Germany; bjoern.schneider@med.uni-rostock.de; 3Research Unit Apoptosis in Hematopoietic Stem Cells, Helmholtz Zentrum München, Feodor-Lynen-Str. 21, 81377 Großhadern, Germany; irmela.jeremias@helmholtz-muenchen.de; 4Comprehensive Cancer Center Mecklenburg-Vorpommern (CCC-MV), Campus Rostock, Ernst-Heydemann-Str. 6, 18057 Rostock, Germany; 5Rudolf-Zenker-Institute for Experimental Surgery, Rostock University Medical Center, Schillingallee 69a, 18057 Rostock, Germany; brigitte.vollmar@med.uni-rostock.de

**Keywords:** acute lymphoblastic leukemia, PDX, biobanking, serial transplantation, engraftment site, proliferation kinetics, mutation profiling, cancer hotspot panel

## Abstract

In acute lymphoblastic leukemia (ALL), conventional cell lines do not recapitulate the clonal diversity and microenvironment. Orthotopic patient-derived xenograft models (PDX) overcome these limitations and mimic the clinical situation, but molecular stability and engraftment patterns have not yet been thoroughly assessed. We herein describe and characterize the PDX generation in NSG mice. In vivo tumor cell proliferation, engraftment and location were monitored by flow cytometry and bioluminescence imaging. Leukemic cells were retransplanted for up to four passages, and comparative analyses of engraftment pattern, cellular morphology and genomic hotspot mutations were conducted. Ninety-four percent of all samples were successfully engrafted, and the xenograft velocity was dependent on the molecular subtype, outcome of the patient and transplantation passage. While *BCR::ABL1* blasts were located in the spleen, *KMT2A*-positive cases had higher frequencies in the bone marrow. Molecular changes appeared in most model systems, with low allele frequency variants lost during primary engraftment. After the initial xenografting, however, the PDX models demonstrated high molecular stability. This protocol for reliable ALL engraftment demonstrates variability in the location and molecular signatures during serial transplantation. Thorough characterization of experimentally used PDX systems is indispensable for the correct analysis and valid data interpretation of preclinical PDX studies.

## 1. Introduction

More than 95% of all therapeutic agents in the oncology sector that demonstrate preclinical activity finally fail during clinical development. This is, in most cases, due to a lack of efficacy [1,2,3]. Preclinical studies are usually performed in in vitro cell culture experiments or using cell line-derived xenograft mouse models. Cancer cell lines have been used extensively to explore the underlying processes of tumor development and initiation, systematic pathway regulation and to identify potential therapeutic targets for anti-tumor applications. Still, cell line-based model systems possess only reduced translational potential; the results of in vitro experiments are hampered because of the limited ability to recapitulate the tumor cell biology, the clonal diversity of cancer samples or the specific tumor cell microenvironment [4,5].

Using patient tumor specimens can significantly reduce the genetic drift and lack of translational significance of cell line models. However, the collection and use of patient-derived primary material is challenging for some tumor entities, depending on the incidence and the sensitivity of the primary material in artificial culture conditions. For hematological malignancies, including acute lymphoblastic leukemia (ALL), sample collection from peripheral blood or excess diagnostic bone marrow aspirates is usually a straightforward process. In vitro cultivation of leukemic cells, however, is extremely challenging and depends on a variety of factors [6]. Several attempts have been made using different media, cytokines or serum conditions, resulting in low success rates and allowing cultivation and proliferation only for days to weeks [7,8,9]. To further replicate the tumor microenvironment, ALL cells can be maintained in coculture with irradiated human or murine stromal feeder cells [10,11,12]. This procedure is technically challenging but allows longer cultivation periods of several weeks up to months.

Patient-derived xenograft (PDX) models have emerged as a superior alternative to overcome the limitations of cell lines and the technical difficulties of primary cell culture [13,14,15]. Primary tumor material is injected or implanted into immunocompromised mice, inducing heterotopic or orthotopic cancer cell proliferation. Leukemic blasts are usually injected intravenously and then migrate to and engraft in the bone marrow and spleen, generating orthotopic PDX models featuring tumor growth within the correct microenvironment. Those models are therefore better suited to mimic the clinical situation, test therapeutic approaches, investigate genetic biomarkers or predict relapse situations [16,17]. Studies have shown that orthotopic ALL PDXs largely resemble the primary tumor’s mutational landscape, as well as gene and protein expression profiles [18,19,20]. A number of experiments in acute myeloid leukemia (AML) have, on the other hand, demonstrated that there is a certain degree of clonal heterogeneity between the primary sample and xenograft, including both gains and losses of alterations [21]. While the initial tumor comprises several subclones, leukemia engraftment often selects for individual cancer cell subsets [22].

Nowadays, the preclinical evaluation of novel therapeutic agents’ efficacy, as well as their safety, toxicity and pharmacological parameters, is carried out using PDX studies. As PDX models can be of predictive value, they can guide a rational clinical trial design, offering a fast phase II-like trial with murine participants of the control and treatment arms being deprived of the same patient sample [23,24]. To cover the vast heterogeneity of ALL and allow statistically significant result interpretation, it is important to include a variety of PDX models covering primary samples of distinct molecular or cytogenetic subtypes, age groups, gender or previous treatments [24]. There are published biobanks of acute leukemias that also include clinical information of the patients. Those repositories mainly collected AML [25,26] or pediatric ALL samples [17,23,27,28,29]. Due to the rarity of adult ALL, there is only limited primary material available for preclinical research [30]; reports of adult ALL PDX models are therefore sparse, and only a few studies provide details on genetic alterations or comparisons between the primary tumor and xenograft material [24,31].

We herein describe the establishment and thorough characterization of an adult ALL biobank containing samples of several cytogenetic backgrounds, as well as orthotopic PDX model generation. We compare engraftment patterns, as well as molecular and cellular properties of the PDX models after up to four passages, and provide rationales for successful xenografting and experiment data handling.

## 2. Materials and Methods

### 2.1. Human Tissue Collection

Blood or bone marrow aspirates were obtained from 44 ALL patients at initial diagnosis or relapse treated at the Rostock University Medical Center, Germany between January 2005 and November 2020 after informed consent. Baseline clinical data (age, sex, subtype, sample source and blast frequency) were provided by Clinic III—Hematology, Oncology, Palliative Medicine of the Rostock University Medical Center.

### 2.2. Tumor Cell Purification and Storage

Peripheral blood or bone marrow aspirates were collected in EDTA or heparin for diagnostic purposes, and the leftover material was anonymized and used for this research within 24 h. Samples were first mixed with an equal amount of warm PBS (PAN-biotech, Aidenbach, Germany) and then separated by density gradient centrifugation (1200× *g*, 12 min, 4 °C, brake 0) using Biocoll separating solution (Merck Millipore, Darmstadt, Germany). Mononuclear cells were taken off and washed in PBS twice. Where necessary, samples were incubated in 10 mL erythrocyte lysis buffer (0.25 mM NH_4_Cl, 10–15 min, 4 °C) and twice washed in PBS. After determination of the cell numbers, backup cell pellets and cell lysates (RIPA lysis buffer, Cell Signaling, Danvers, MA, USA) were stored at −80 °C for subsequent DNA, RNA or protein analyses. The tumor cell frequency was determined by flow cytometry (see below). Viable cells (5–10 × 10^6^ cells per vial) were frozen in freezing medium containing 10% DMSO (Sigma-Aldrich, St. Louis, MO, USA) and 90% cold FCS (PAN-biotech) and stored in liquid nitrogen until further use.

### 2.3. Sample Preparation for Xenotransplantation

Cells were thawed at 37 °C, carefully transferred into 2 mL prewarmed sterile FCS and incubated for 1 min. 23 mL of prewarmed RPMI 1640 medium (PAN-biotech) were then added dropwise, carefully mixed and incubated for 1 min before centrifugation at 180× *g* for 8 min at 20 °C. The supernatant was discarded, and the cells were carefully resuspended in 10 mL cold PBS containing 2% FCS before centrifugation at 180× *g* for 8 min at 10 °C. After discarding the supernatant, the cells were resuspended in cold PBS/2% FCS, and the cell numbers were determined. For xenotransplantation, the cells were prepared in PBS in a density of 2.5 × 10^7^ per mL.

### 2.4. Orthotopic Xenografting of Tumor Cells

Mice were bred and housed in the accredited laboratory animal core facility of the Rostock University Medical Center with access to water and standard chow *ad libitum*. All experiments were carried out in a laboratory setting, and no intervention was performed within the animal housing and breeding rooms. Experiments were approved by the review board of the federal state Mecklenburg-Vorpommern, Germany (reference numbers: LALLF MV/7221.3-1.1-002/15 and 7221.3-1.1-063/20). For xenotransplantation, cells were thawed and prepared in PBS at a density of 2.5 × 10^7^ per mL. Depending on the available cell numbers, 2.5 × 10^5^ to 3.7 × 10^6^ tumor cells were intravenously injected into up to four 8–16-week-old NOD.Cg-*Prkdc^scid^ Il2rg^tm1Wjl^*/SzJ (NSG, The Jackson Laboratory, Bar Harbor, ME, USA) mice for first-generation engraftment. For the following passages, 2.5 × 10^6^ cells per mouse were injected when sufficient cell numbers were available. The total number of injected animals and the respective cell counts are summarized in Appendix A. The animals’ weight and wellbeing were checked daily and documented on score sheets.

### 2.5. Monitoring of Tumor Cell Proliferation

Peripheral blood tumor cell frequencies were routinely evaluated by flow cytometry every two to four weeks with shorter intervals upon tumor cell engraftment (see below). Blood count analyses were performed once or twice weekly using the ADVIA 2120 Hematology System (Siemens, Erlangen, Germany) with their respective controls. After two rounds of serial transplantation, the cells of two patients (#0122 and #0159) were stably transduced with enhanced firefly luciferase subcloned into the pCDH-EF1-MCS-T2A-copGFP vector (System Biosciences, Mountain View, CA, USA) [32]. This modification allowed for tumor cell detection by longitudinal in vivo bioluminescence imaging. Mice were injected with 4.5 mg D-luciferin (Goldbiotechnology, St. Louis, MO, USA) and narcotized for imaging using the NightOWL LB983 imaging system (Berthold Technologies, Bad Wildbach, Germany) in the ventral and dorsal positions.

### 2.6. Tissue Collection and Sterile Preparation

Once the blast frequencies reached 30% in the peripheral blood or the mice met predefined humane endpoints, the experiment was terminated, and the mice were euthanized. Spleens were weighed and measured and stored in ice-cold RPMI 1640 medium containing 20% FCS until further processing. Spleens were sterilely passed through a 100 µm cell strainer and washed in PBS at 180× *g* for 8 min. Cells were then suspended in 1 mL PBS, and 9 mL cold erythrocyte lysis buffer (10 mM KHCO_3_, 155 mM NH_4_Cl and 0.1 mM EDTA) were added and incubated at 4 °C for 15 min. After centrifugation, cells were washed twice in 20 mL cold PBS before the cell numbers were determined. Femurs and tibiae were collected and stored in ice-cold RPMI 1640 medium containing 20% FCS until further processing. Bones were opened under sterile conditions, and the bone marrow cells were flushed out with PBS/2% FCS using 26G and 27G syringes, respectively. Erythrocyte lysis was carried out if necessary. After centrifugation, cells were washed twice in 20 mL cold PBS before cell numbers were determined.

### 2.7. Preparation and Storage of Backup Sampls

For viable freezing of native cells, up to 1 × 10^7^ cells per vial were centrifuged (180× *g*, 8 min), suspended in 900 µL FCS and distributed into NUNC cryotubes (Thermo Fisher Scientific, Waltham, MA, USA) containing 100 µL DMSO. Tubes were placed in a Mr. Frosty™ freezing container (Thermo Fisher Scientific) and stored at −80 °C for at least 24 h before transferring the samples into liquid nitrogen. For genetic and expression analyses, bone marrow and spleen cells were centrifuged at 2000× *g* for 5 min and stored at −80 °C as cell pellet (1.5–5 × 10^6^ cells), RNA lysate in QIAzol lysis reagent (Qiagen, Hilden, Germany) (1.5–5 × 10^6^ cells) or protein lysate in RIPA buffer (Cell Signaling) containing a protease/phosphatase inhibitor cocktail (Cell Signaling) (4–8 × 10^6^ cells).

### 2.8. Determination of Tumor Cell Frequency by Flow Cytometry

For flow cytometry-based evaluation of the tumor cell frequency in the bone marrow and spleen, 1 × 10^5^–1 × 10^6^ cells in 100 µL were stained with anti-human CD45-FITC (clone 2D1) and anti-human CD19-PE (clone 4G7; both Becton Dickinson, Heidelberg, Germany) at 4 °C for 15 min and washed twice (180× *g*, 8 min) in cold PBS before measuring using FACSCalibur or FACSVerse and CellQuest Pro or FACSuite software tools (all Beckton Dickinson), respectively. For determination of the peripheral blood blast frequencies, 20–60 µL of blood were collected from the tail vein and stained as mentioned above. Subsequently, erythrocytes were lysed during 15 min of incubation with 2 mL erythrocyte lysis buffer. Cells were washed twice in 2 mL PBS and analyzed by flow cytometry.

### 2.9. Cytospin Preparation and Pappenheim Staining

For morphological analyses, 5 × 10^4^ cells were suspended in 200 µL PBS and spun onto microscopic slides for 10 min at 700 rpm (Shandon Cytospin 3, Thermo Fisher Scientific). Slides were then stained in May-Grünwald solution (Merck, Darmstadt, Germany) for 6 min, followed by three washing steps in washing buffer (pH 7.2; Merck) for 1 min. The subsequent staining in 1:10 diluted Giemsa solution (Merck) was followed by three washing steps before rinsing the slides in distilled water for 5 s. Images were taken at 100-fold magnification using the EVOS™ XL core imaging system (Thermo Fisher Scientific).

### 2.10. Short Tandem Repeat (STR) Fingerprint Analysis

Tumor cell DNA was isolated from cell pellets using the NucleoSpin^®^ Tissue kit (Macherey-Nagel, Düren, Germany) according to the manufacturer’s instructions. STR fingerprint analysis was carried out using the STR multiplex PowerPlex™ 1.1 kit (Promega, Madison, WI, USA), analyzing eight STR loci (vWA, TH01, TPOX, CSF1, D5S818, D13S317, D7S820 and D16S539). The PCR was performed in a C1000™ thermal cycler (Bio-Rad, Hercules, CA, USA) in a total volume of 12.5 µL containing 25 ng gDNA and 0.8 mM dNTP mix (Bioron, Ludwigshafen, Germany), 0.5 U DFS-Taq (Bioron) and 3.6 µM primer mix using the following protocol: 2 min initial denaturation at 96 °C, 30 cycles of 30 s denaturation at 94 °C, 2 min annealing at 59 °C and 90 s elongation at 72 °C, followed by a terminal elongation of 45 min at 60 °C. The generated fragments were separated by capillary electrophoresis (ABI 3500 Genetic Analyzer, Applied Biosystems, Waltham, MA, USA) in 96-well Multiply^®^ half-skirt plates (Sarstedt AG, Sarstedt, Germany) containing 2 µL of PCR amplicon, 22.7 µL Hi-Di™ formamide (Thermo Fisher Scientific) and 0.3 µL internal size standard (Gene Scan™ 500 LIZ^®^, Applied Biosystems). GeneMapper^®^ software (Applied Biosystems, version 4.1) was used to analyze the electropherograms. For the FAM™ and HEX™-labeled primers, the peak cutoff was defined at 15% and 33.3% of the highest peak, respectively. Allelic ladders were used for allele designation.

### 2.11. Analysis of Genomic Hotspot Single Nucleotide Variants

DNA of the primary patient material PDX samples was analyzed using the Ion AmpliSeq™ Cancer Hotspot Panel v2 on an Ion PGM system according to the manufacturer’s guidelines. Sequencing was performed with a minimum of 500 reads base coverage. Variant calling and annotation was performed using the Variant caller software plug-in and the IonReporter platform (all Thermo Fisher Scientific) according to the routine diagnostic workflow.

### 2.12. Statistical Analysis

All values were expressed as the mean ± standard deviation. Gaussian normality distribution was tested in all cases, determining the following parametric or nonparametric post hoc test. The exact tests are indicated in the respective figure legends. Statistical analyses were performed using GraphPad PRISM software (version 8). Statistical significance was defined as * *p* < 0.05, ** *p* < 0.005 and *** *p* < 0.001.

## 3. Results

### 3.1. Characterization of the Patient Cohort

Our sample cohort comprised 44 individual adult ALL samples collected between 2006 and 2021 at the Rostock University Medical Center (Table 1). The samples were equally distributed between males and females, and the patients were between 18 and 84 years old (median 50.5 years). Thirty-seven patients were enrolled at their initial ALL diagnosis, and seven patients were recruited when they presented with relapsed disease. Most cases (40/44) were B-cell neoplasia, with three samples described as biphenotypic. Most patients harbored cytogenetic aberrations, with *KMT2A* translocations (seven patients) and *BCR::ABL1* translocations (fourteen patients) occurring most frequently (Appendix A). *KMT2A* translocations were mostly detected in pro B-ALL, while *BCR::ABL1* aberrations occurred mainly in common B-ALL. Deletion of the p16*^INK4A^* locus was reported in four cases. Twelve patients were still alive at the time of data analysis, while seventeen patients had died. For the fifteen cases with the last contact longer than one year ago, the last follow-up date was plotted, and their status was referred to as “unknown” (Appendix A). The material used for the subsequent analysis, tumor cell isolation and xenografting was bone marrow in twenty cases and peripheral blood in twenty-one cases (three unknown).

Eleven B-ALL patients were further selected for detailed genomic analyses of cancer hotspot regions based on their cytogenetic subtype and sample availability (Appendix A). A total of fourteen different amino acid-changing alterations were detected. Half of the mutations were considered pathogenic, likely pathogenic or classified as uncertain by the ClinVar database [33]. Benign or likely benign alterations were more likely to be homozygous or present in roughly 50% of the alleles, arguing for germline variants. Pathogenic and previously undescribed mutations, on the other hand, were frequently detected with low allele frequencies. Only four patients (#0122, #0094, #0134 and #0159) shared the seven pathogenic and uncertain mutations and also had the highest overall number in changes. One or two variants (*n* = 3 and 4, respectively) were identified in most samples. Patient #0159 exhibited the most mutations, with a total number of five alterations. The drug response-mediating *TP53* Pro72Arg polymorphism was the most common and present in all but one sample. Two other *TP53* variants were detected in patients #0094 and #0181. Tyrosine kinases like *MET* and *FLT3* were also affected in multiple samples. Ras signaling genes were altered in three patients, with one patient sharing *KRAS* and *FLT3* mutations (#0122) and another *NRAS* and *PIK3CA* alterations (#0159).

### 3.2. Establishment of Patient Derived Xenograft Models

From the initial 44 patients-containing ALL cohort, eighteen samples were selected for orthotopic in vivo engraftment (Table 1) based on their molecular subtype, survival parameters and cytogenetic aberrations to include a variety of cases in the xenograft studies.

The primary samples were injected in a total of 31 mice for the first passage of tumor cell expansion. We first evaluated the parameters that might influence tumor cell engraftment and proliferation. Flow cytometric detection of human blasts (CD45^+^/CD19^+^ for B-ALL and CD45^+^/CD5^+^ or CD45^+^/CD7^+^ for T-ALL, based on surface marker phenotyping in the original tumor material) showed engraftment in all but one (patient #0031) patient sample, resulting in an engraftment rate of 94.4%. Two female mice were injected with 1.1 × 10^6^ B-ALL patient #0031 cells harboring an IGH locus rearrangement and closely monitored for 191 days. Regular blood count analyses and blast determination by flow cytometry did not indicate any signs of leukemic burden. Both mice gained weight during the procedure, and their spleens were not enlarged at experiment termination.

Overall, tumor cell injections in a total of 101 mice were well-tolerated by all animals, and leukemia engraftment did not result in significant weight reduction (Figure 1A). No signs of illness (ruffed fur, sunken eyes, reduced activity and apathetic behavior) were detected upon experiment termination. Regular blood count analyses of 58 representative animals (Appendix A) revealed significant changes in several blood parameters after leukemic blast engraftment compared to the basal levels. While the leukocyte counts increased by 82%, the erythrocyte and thrombocyte numbers dropped by 18% and 27%, respectively. The hemoglobin and hematocrit values were also reduced by 17% and 15%. The proportion of basophils significantly increased, while the percentage of monocytes decreased. Simultaneously, the total number of neutrophil, eosinophil and basophil granulocytes was raised. The lymphocyte cell counts also increased significantly.

Mice exhibiting leukemic engraftment were sacrificed when the peripheral blood blast frequency exceeded 30%. This point was reached earlier, albeit not significantly, when the survival of the patient after sample collection was shorter than one year (Figure 1B). On the other hand, patient gender, patient age and sample origin did not influence the engraftment velocity (Figure 1C–E). The molecular subtype of the primary tumor played a certain role in the engraftment velocity, with mice xenografted with patient samples harboring a *BCR::ABL1* translocation demonstrating the lowest mean time to experiment termination (Figure 1F). Tumors with *KMT2A* rearrangements or normal karyotypes had slightly longer engraftment periods. Expansion of a sample with (1)(q23) and (12)(p13) deletion, as well as trisomy 21 (patient #0074) and T-ALL cells, was slower and took at least twice as much time as in B-ALL samples. The quickest overall engraftment was achieved by patient #0122 harboring an *KMT2A* translocation. This data, however, are only representative for the given sample collection and remain to be validated, as only two T-ALL patient samples and three samples with normal karyotypes were included in the study cohort. Besides the molecular subtype, the number of injected tumor cells is of importance for the engraftment velocity (Figure 1G). The injected blast number negatively correlated with the expansion time, demonstrating that the blast frequency threshold of 30% in peripheral blood was achieved significantly faster when a higher amount of cells was injected. Other technical points like the gender of the engrafted mouse did not influence the speed of engraftment (Figure 1H).

We next evaluated if the distinct molecular subtypes exhibited different engraftment patterns (Figure 2). Indeed, differences in bone marrow and spleen infiltration were observed (Figure 2A,B). However, the sample numbers for T-ALL, other cytogenetic subtypes and specimens without cytogenetic alterations were very small and should be considered with great care. The mean bone marrow blast frequency was highest in the samples with *KMT2A* translocations (Figure 2A). Comparable values were achieved by samples with other cytogenetic aberrations or of T-ALL origin, while *BCR::ABL1*-mutated tumors had lower blast frequencies (not significant). Cells without any chromosomal conspicuity had the lowest mean bone marrow infiltration. In contrast, *BCR::ABL1*-positive cases and those samples without cytogenetic aberrations demonstrated the highest overall blast frequency in the spleen (Figure 2B). The mean spleen infiltration in the T-ALL samples was only around half as high as in the other subtypes and significantly reduced compared to all the other subtypes. Interestingly, the spleen weight of the samples with *BCR::ABL1* and *KMT2A* rearrangements, which had rather high spleen infiltrations, was lower than the weight of the spleens explanted from the T-ALL-engrafted animals (Figure 2C). Taken together, for all mice irrespective of their molecular subtype the spleen blast frequency was weakly positively correlated with the spleen weight (Figure 2D). A significant correlation between spleen and bone marrow infiltration was calculated for the *KMT2A* rearranged samples and cases without chromosomal aberrations (Figure 2E).

### 3.3. Characterization of PDX Models throughout Serial Transplantation

Throughout the process of xenografting, the tumor cell proliferation and location were thoroughly monitored using in vivo bioluminescence imaging (BLI) and the flow cytometry-based evaluation of circulating blast frequencies. Comparing both techniques, BLI was by far more sensitive in detecting tumor cell engraftment, generating positive results around ten days earlier (Figure 3A,B). Flow cytometry resembled the growth kinetics detected by BLI without the technical saturation observed in imaging, suggesting that flow cytometry is superior for the longitudinal evaluation of tumor progression at later stages. The primary endpoint of the xenograft experiments was a tumor cell frequency of >30% in peripheral blood. As the BLI signal was already saturated at this stage of the disease, we decided to monitor the tumor progression via flow cytometry in consecutive experiments.

We next evaluated if the number of serial transplantation events was of relevance for the velocity of tumor cell expansion (Figure 3C). Detailed engraftment analyses of 13 patient samples (five *BCR::ABL1* translocations, six *KMT2A* rearrangements and two T-ALLs) revealed ambiguous patterns with the time to experiment termination reduced and prolonged in eight and four cases, respectively. Especially *KMT2A*-positive samples were heterogenous, with half of the evaluated cases growing faster and the other half slower in the following passages. A clear trend towards faster tumor cell engraftment and proliferation was obvious in *BCR::ABL1*-mutated samples (four out of five), as well as T-ALLs.

In general, the time between the first detection of leukemic blasts in the peripheral blood significantly correlated with the time between the initial tumor cell detection and the time to experiment termination, allowing conclusions about proliferation kinetics (Appendix A). When comparing all animals irrespective of their molecular subtype, the mean expansion time decreased from passage 1 to passage 2 and further dropped in passage 3 (Appendix A). It is to be noted, however, that mice with faster engraftment were selected for further cell transplantation in the next serial passage, possibly accounting for an observation bias. We thus subsequently analyzed only the engraftment velocity of animals involved in serial transplantation experiments and excluded all mice whose cells were not further used for injection in the next passage. Matching our previous results, these analyses also revealed moderately accelerated tumor cell engraftment, ruling out the above-mentioned error source.

Examining the location of tumor cell engraftment, we next analyzed the blast frequency in the bone marrow and spleen (Figure 3D). While *KMT2A*-rearranged samples, as well as T-ALLs, kept a pattern of higher bone marrow infiltration throughout all the passages, the *BCR::ABL1*-mutated models lost their characteristic feature of higher blast frequencies in the spleen after the second and third serial transplantation rounds. While the infiltration of the spleen remained constant, the bone marrow blast frequencies increased with each round of passaging.

We next sought to investigate the effect of serial transplantation on the tumor cells themselves. Looking into the cells’ morphology, different patterns and grades of size, vacuolization, the nucleus-to-cytoplasm ratio and chromatin condensation were present in the distinct patients (Appendix A). Serial transplantation did not affect those properties. On the molecular level, short tandem repeat (STR) fingerprinting was used to assess the changes in genetic stability. Six primary *KMT2A*-positive patients’ samples and up to four consecutive samples after serial xenografting were evaluated by this method. Except for one patient (#0134), no shifts were detected. In patient #0134, however, STR markers vWA and D5D181 showed an altered pattern in passage 3 (Appendix A). This change was accompanied by an increase of nuclear homogeneity and reduction of vacuolization (Appendix A). Sequencing of cancer-related hotspot regions revealed the rise of a *TP53* c.215C>G mutation at the same time (Appendix A).

Throughout serial transplantation, a total of 36 different alterations affecting 23 genes were observed. The most common variants were *APC* c.4479G>A and *HRAS* c.81T>C, which were present in four samples; *FLT3* c.1310-3T>C, *KDR* c.798+54G>A and *RET* c.2307G>T, which were detected in five samples and *CSF1R* c.*36A>TC, *EGFR* c.2361G>A, *PDGFRA* c.1701A>G and *TP53* c.215C>G, which were present in all six patients. None of those variants induced amino acid changes. All those changes are known germline alterations and presented either homozygous or heterozygous in our cohort. The *FLT3* c.1775T>C mutation was the only proven somatic and pathogenic mutation that was present in more than one sample (patients #0094 and #0134). Table 2 documents all the variants observed.

The variant analysis further revealed changes during serial transplantation in all but one (patient #0152) sample (Figure 4 and Appendix A). Relevantly altered allele frequencies occurred in zero (patient #0152; 0/12 variants) to 28.6% (patient #0122; 4/14 variants) of all variants (mean 11.6%). While the loss of allelic variants occurred mainly between the primary sample and first passage, additional changes or increased allele frequencies were only detected after at least two rounds of serial transplantation. For two patients, cells of two individual mice injected with the same sample were compared (passage 2 of patients #0122 and #0134). Variant frequencies of one mouse resembled the second for patient #0134. In patient #0122, however, mouse 2 harbored a heterozygous *FGFR3* c.1953G>A variant that was not present in mouse 1 but was passed on to the next passage.

Considering only changes after the initial round of engraftment, four out of six PDX models demonstrated stable molecular profiles. The patient #0134-derived model exhibited a slight increase of a *TP53* polymorphism after two rounds of xenotransplantation. Following the initial engraftment, patient #0122-derived xenograft mice of the second passage showed a reduced *KRAS* mutation allele frequency, as well as a novel *FGFR3* variant. Overall, relevant changes occurred in only 3.6% of all the detected variants after the first round of xenotransplantation.

Filtering out the germline mutations and only focusing on known somatic or probably somatic alterations, seven out of twelve detected variants underwent major changes in the allele frequency (Figure 4). Six changes accounted for the clones present in the primary sample that were lost during the initial xenografting. During serial transplantation following the initial engraftment step, no further changes occurred in the somatic variants except for a drop, but not loss, of a *KRAS* mutation in case #0122.

## 4. Discussion

PDX models are versatile and valuable model systems to investigate ALL biology, recapitulate leukemogenesis and interrogate the therapeutic response. We herein describe the process of ALL biobank establishment and characterization and highlight different engraftment patterns and kinetics in distinct molecular subtypes. Previous reports of ALL PDX cohorts focused on childhood cases [17,23,27,28,29], and with limited cases and availability of adult ALL samples, this report represents a unique collection of well-characterized primary and xenograft cells. Compared to other studies, our protocol results in increased engraftment rates with successful tumor cell proliferation and conserved immunophenotype in seventeen out of eighteen cases (94%). For ALL and AML, other groups reported engraftment rates between 33% and 93% [17,24,27] and between 59% and 88% [21,25,26], respectively. Serial transplantation for up to four passages was successful for all cases attempted. Patel et al. reported the requirement of total body irradiation before cell injection of non-MLL rearranged cases in their cohort of seventeen adult ALL samples, achieving an engraftment rate of 76.5% [34]. In our hands, no preconditioning was necessary for the successful proliferation of any cytogenetic subgroup.

Adult ALL is a heterogeneous disease comprising several molecular subtypes linked to varying prognoses. Previous reports noted that more aggressive hematological neoplasms were easier to xenograft and amplified faster than cells with low-risk aberrations [17,21,26,27]. This matches our findings of the fastest mean engraftment times in high-risk cohorts with *BCR::ABL1* or *KMT2A* translocations, as well as patients who died within twelve months of sample collection. In contrast, the sample origin (bone marrow or peripheral blood) had no influence on the engraftment rate, matching previous results [31]. We herein, for the first time, discuss subtype-specific engraftment patterns in more detail, elucidating different engraftment sites and proliferation kinetics using flow cytometry and in vivo bioluminescence imaging. The latter technique further enables therapeutic monitoring, as well as blast localization even at very low tumor cell levels, offering the tracking of minimal residual disease and engraftment sites [35,36]. We observed significant differences between the molecular entities, with *KMT2A*-positive and T-ALL samples engrafting predominantly in the bone marrow while *BCR::ABL1* cases manifested in the spleen tissue. This matches the findings of Aoki et al., who found higher mean blast frequencies in the bone marrow compared to the spleens of 15 *KMT2A-AF4* and seven *KMT2A-ENL*-rearranged ALL cases [37]. For T-ALL, previous experiments also observed increased bone marrow manifestation, supporting our findings, which are, however, limited to a very small cohort of two different T-ALL primary samples amplified in a total of ten animals [38,39]. In line, a study using a ^111^In-labeled CD19 antibody found higher spleen, compared to bone marrow, infiltration rates in *BCR::ABL1* xenografted animals [40]. Interestingly, *BCR::ABL1*-rearranged samples seemed to lose this pattern during serial transplantation due to increased engraftment in the bone marrow.

At the same time, the expansion times decreased significantly after serial transplantation of all the cytogenetic subtypes, especially in those samples that showed very long expansion times during the first round of engraftment. However, a few cases showed opposite results. This matches the findings of Belderbos et al., who also observed ambiguous kinetics after B-ALL serial transplantation [27]. In contrast, no relevant differences in engraftment velocity between the B-ALL xenograft passages was observed in the study of Gopalakrishnapillai et al. [41]. For our specimens, delayed engraftment might, in some cases, be a result of technical difficulties like insufficient i.v. injection or low leukemic blast viability. On the other hand, a higher tumor cell purity compared to the initial primary sample could explain shorter expansion times. Clonal selection of fast proliferating leukemic cells is another likely explanation of accelerated proliferation kinetics [27,31].

We observed molecular signatures of primary samples and serially transplanted xenografts using Ion AmpliSeq™ Cancer Hotspot Panel v2. Of the six samples selected for detailed analyses, five displayed relevant mutational changes throughout xenotransplantation. This matches previous findings for AML and solid cancer samples [21,42]. It is to be noted, however, that the used Cancer Hotspot Panel v2 only comprises a subset of cancer-relevant genetic alterations and does not provide information on copy number variations or detailed shifts in tumor sample clonality. Focusing on a single endpoint (30% blasts in peripheral blood) further displays a relatively late stage of tumor progression while earlier observation time points could shed light on distinct subclones.

The clinical and therapeutical impacts of the observed changes between the primary sample and initial PDX model are difficult to estimate. Using single-cell sequencing to analyze the clonal architecture of primary and relapse samples, as well as the corresponding PDX models, studies in AML and multiple myeloma have demonstrated that similar clones outgrow in relapsed patients and PDX samples [43,44]. These findings suggest that the selective pressure of xenografting resembles the processes of therapeutic intervention and implicate that PDX models are a suitable tool to mimic the clinical relapse situation.

In line with recent studies, we observed a stable transmission of variants from primary sample to xenografts throughout serial transplantation for most cases (71/82 variants; 86.6%) [18,21,24,31]. Interestingly, and demonstrating great molecular stability of the established PDX models, only very few additional changes occurred after the initial engraftment. With 96.4% of all detected variants, and including all but one somatic change, remaining at their basal allele frequency level of passage 1, our model systems promise high reliability and mutational continuity.

Except for one *TP53* mutation in patient #0094, a loss of variants occurred solely in cases with minor allele frequencies between 2% and 6%, and variants were lost during the first engraftment round. This is probably due to the clonal selection of certain subgroups of leukemia-initiating cells and matches the findings of Richter-Pechanska et al., as well as other studies [18,27,31]. While the loss of or decreased mutation frequency were observed in almost 10% of all the variants detected, and in contrast to a study in childhood ALL [28], a gain of a mutation was only observed in one case. Patient #0122 obtained an intronic *FGFR3* variant after two rounds of transplantation. Two recent reports described this phenomenon and postulated that rising mutations during xenotransplantation might not be detected in the primary sample; however, they are most likely already present as a founder clone at a very low allele frequency [18,45]. Interestingly, the described *FGFR3* variant was detected in only one out of two mice injected in parallel with the same cell suspension. Mice retransplanted with the variant-containing cells maintained the mutation at a comparable allele frequency, ruling out a technical problem. Comparative molecular analyses of two mice xenografted with the same cell suspension were performed for two patients (#0122 and #0134). The above-mentioned *FGFR3* mutation cooccurred with a reduced *KRAS* allele frequency in patient #0122. All the other mutations, as well as all variants in the patient #0134 samples, were comparable between the respective mice of the same passage, confirming the results of Vick et al. [25], as well as Wang et al. [31]. The polyclonal structure of the underlying cell suspension is most likely the reason for the outgrowth of different clones in two mice injected with the same sample material.

## 5. Conclusions

As the classical cell line models do not recapitulate the polyclonal landscape of the initial tumors and primary cells are very difficult to cultivate in vitro, PDX systems are indispensable for the evaluation of novel therapeutic approaches and other hematological issues. Although there are obvious changes in the molecular landscape between primary sample and PDX mice in the majority of the cases investigated, PDX model systems remain a crucial part of preclinical ALL research. Our data demonstrated that PDX models, once stably established, maintain their mutational landscape throughout several passages. There are, however, important questions and topics that need to be considered when working with xenograft models. First of all, it is of fundamental importance that the growth kinetics, molecular signature and stability are characterized for the sample intended to use. The correct PDX model must then be carefully chosen for the desired research question, including the selection of a suitable host mouse strain. Finally, when analyzing the impact of certain molecular markers on tumorigenesis or therapeutic responses, the presence and maintenance of the aberration, as well as the allele frequency, must be regularly characterized to avoid misinterpretation of the obtained in vivo data. Given the thorough knowledge of the system’s properties, PDX models are a powerful and valuable tool in preclinical and even clinical research, enabling the rational design of phase II-like studies in a murine setting [15,23,24].

## Figures and Tables

**Figure 1 cells-11-00150-f001:**
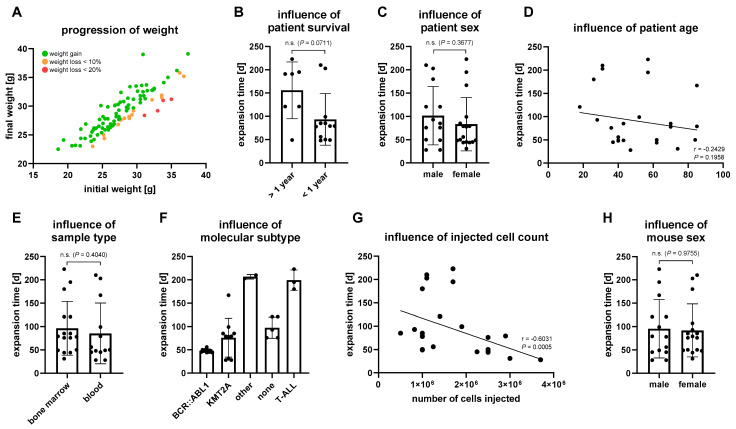
Engraftment-influencing parameters. Every dot represents a separate animal. (**A**) Comparison of the initial weight and weight at experiment termination of all 101 mice used for tumor cell expansion. Weight gain is indicated by green dots, while weight loss up to 10% and 20% is displayed in yellow and red, respectively. (**B**–**H**) Influence of clinical–pathological parameters on tumor cell engraftment and proliferation in the first PDX passage. (**B**) Only patients with confirmed date of death or a recent checkup no older than twelve months were included in the analysis to investigate the influence of patient survival. Samples of unknown status or when the last contact was longer than one year ago were excluded. *n* = 19 mice, mean ± standard deviation, Mann–Whitney test. (**C**) Influence of patient sex on engraftment velocity. *n* = 30 mice, mean ± standard deviation, Mann–Whitney test. (**D**) Correlation of patient age at sample collection with engraftment kinetics. *n* = 30 mice (Spearman’s correlation coefficient r). (**E**) Influence of sample origin (bone marrow or peripheral blood) on engraftment velocity. *n* = 29 mice, mean ± standard deviation, Mann–Whitney test. (**F**) Influence of the molecular subtype of the primary tumor on engraftment speed. Patients with *BCR::ABL1* or *KMT2A* translocations and additional aberrations were only considered for the *BCR::ABL1* and *KMT2A* cohorts, respectively. *n* = 30 mice, mean ± standard deviation, no statistical evaluation due to limited sample numbers in subgroups. (**G**) Correlation of the number of cells injected and the engraftment velocity. *n* = 30 mice (Spearman’s correlation coefficient r). (**H**) Influence of mouse sex on engraftment speed. *n* = 30 mice, mean ± standard deviation, Mann–Whitney test.

**Figure 2 cells-11-00150-f002:**
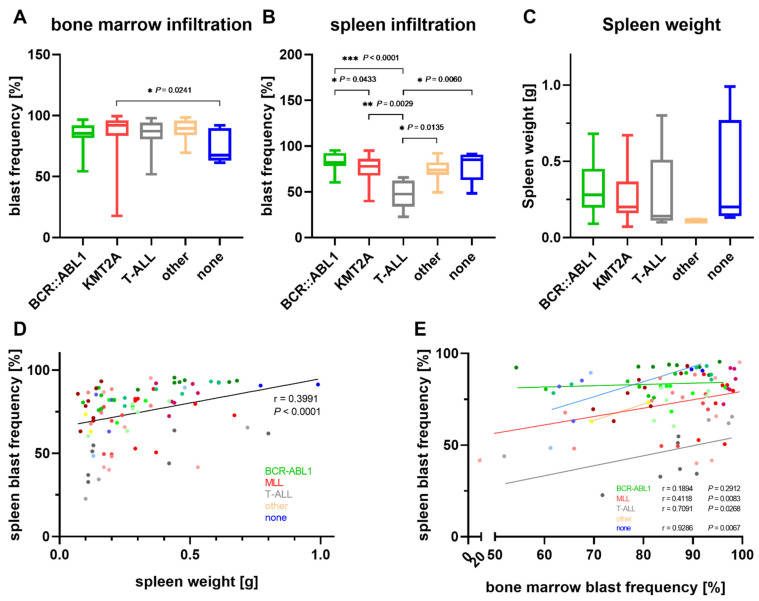
Influence of the molecular subtype on the engraftment sites. All 101 animals used for cell expansion irrespective of the engraftment passage were considered for these analyses. (**A**,**B**) Bone marrow and spleen infiltration were determined by a flow cytometric analysis of human C45^+^/CD19^+^, C45^+^/CD5^+^ and CD45^+^/CD7^+^ blasts isolated from animals at the end of the expansion experiment. Kruskal–Wallis and post-hoc Dunn’s multiple comparisons test. (**C**) Influence of the molecular subtype on the spleen weight. Kruskal–Wallis test. (**D**,**E**) Correlation analysis between the spleen weight and spleen infiltration (**D**) and spleen and bone marrow blast frequency (**E**). Each dot represents a single animal, and mice engrafted with the same patient material are displayed in the same color. Similar color shades indicate the same molecular subtype. Spearman’s correlation value r.

**Figure 3 cells-11-00150-f003:**
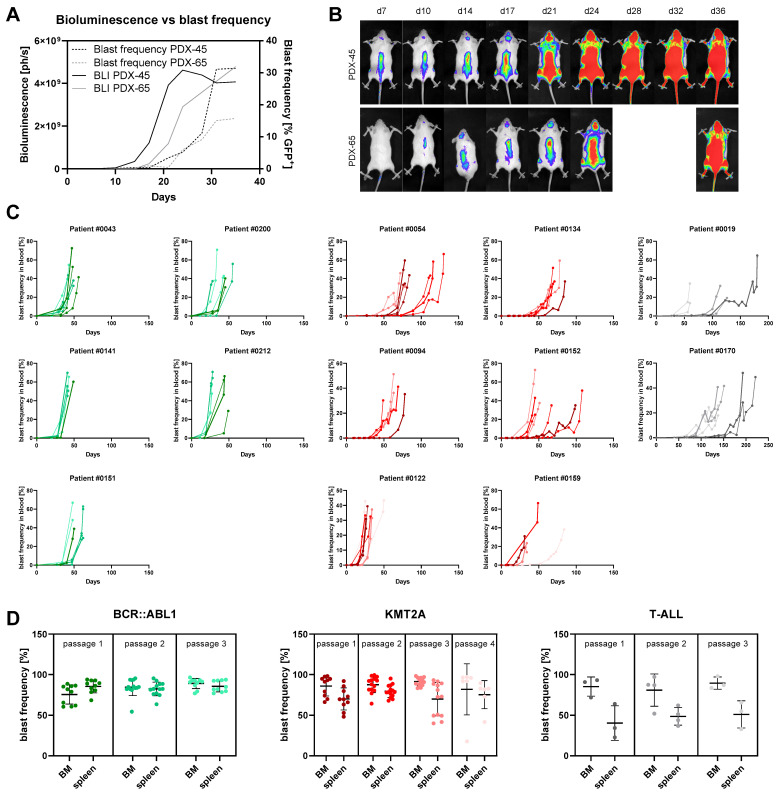
Evaluation of tumor engraftment and growth kinetics. (**A**) Longitudinal quantification of the bioluminescence (full lines) and circulating blast frequency (dotted lines) of two representative animals. (**B**) Corresponding bioluminescence images of mice depicted in Figure 3A. (**C**) Influence of serial transplantation on the engraftment velocity. Samples with *BCR::ABL1*, *KMT2A* and T-ALL are painted in green, red and grey, respectively. Flow cytometric determination of the tumor cell frequency in peripheral blood was performed throughout the observation period. Each line represents an individual animal, and mice engrafted within the same passage are displayed in the same color. Earlier passages are depicted in darker colors. A line printed in bold indicates that the following passage was derived from the indicated animal. (**D**) Flow cytometric analysis of bone marrow (BM) and spleen infiltration upon termination of the experiment. Each dot represents an individual mouse. Mean ± standard deviation.

**Figure 4 cells-11-00150-f004:**
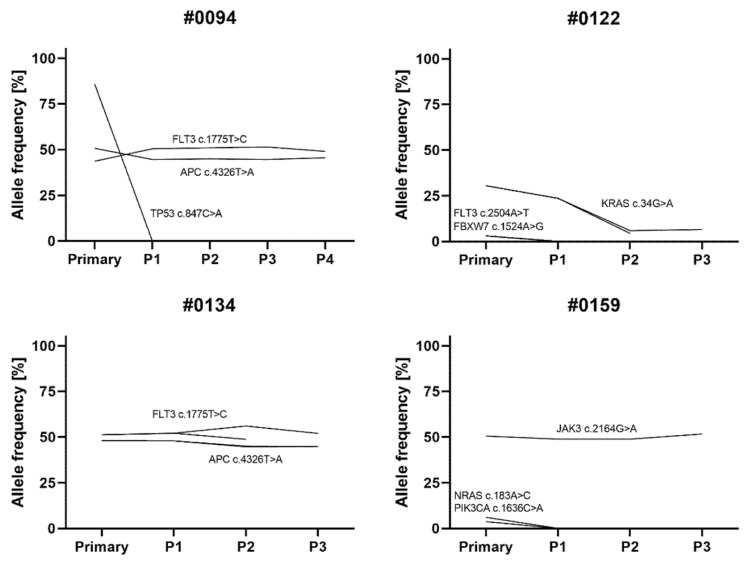
Presence of single nucleotide variants in the *KMT2A*-positive primary samples and consecutive PDX passages. Only somatic or likely somatic variants detected using the Ion AmpliSeq™ Cancer Hotspot Panel v2 (Thermo Fisher Scientific) are summarized irrespectively of their pathogenicity. Two individual mice were analyzed in passage 2 of patients #0122 and #0134 and depicted by separate lines. Only cells from one of those mice were used for the subsequent xenografting to generate passage 3. P, passage.

**Table 1 cells-11-00150-t001:** Clinical parameters and molecular characteristics of the ALL cases included in the study.

Laboratory ID	Subtype	Sex	Age at Sample Collection	Diagnosis	Age at Diagnosis	Sample Type	Cytogenetic Aberrations				Survival		Xeno
							Ph+	MLL	p16	Other	Condition	Days Since dx	
0054	pro B-ALL	f	70	initial	70	bone marrow					dead	358	✓
0054	pro B-ALL	f	71	relapse	70	blood					dead	358	
0072	pro B-ALL	m	63	initial	63	blood				near tetraploid	dead	77	
0122	pro B-ALL	m	47	initial	47	blood				+X, −, +21	unknown	2018	✓
0130	pro B-ALL	f	18	initial	18	bone marrow				del(12)(p13)	dead	160	
0134	pro B-ALL	m	43	initial	43	bone marrow				+22	dead	205	✓
0152	pro B-ALL, biphenotypic	f	52	initial	52	blood					dead	66	✓
0210	pro B-ALL, biphenotypic	m	67	initial	67	bone marrow				del(6)(p21p25), ins(2;6)(p22;?p22?p23?), r(6)(p12q13), dic(16;17)(q11;p11)	alive	237	
0012	common B-ALL	f	77	relapse	76	blood					dead	285	
0031	common B-ALL	f	37	initial	37	unknown				IGH rearrangement	dead	1186	X
0043	common B-ALL	f	40	initial	40	blood					dead	81	✓
0062	common B-ALL	m	21	initial	21	blood				del(6)(q?16q23), +19, add(20)(p)	alive	4155	
0065	common B-ALL	f	20	initial	20	blood					unknown	4	
0071	common B-ALL	m	73	initial	73	blood					dead	12	
0082	common B-ALL	m	63	initial	63	bone marrow				inv(3)(p21;q26), der(7;8)(q10;q10), del(7)(q22), del(13)(q14q31), −14, −15, −Y	alive	3713	
0100	common B-ALL	m	42	initial	42	bone marrow				+der(5), −22, +der(22)x2	alive	3457	
0125	common B-ALL	f	75	initial	75	blood					unknown	-	
0147	common B-ALL	f	55	initial	55	blood				+6, +8, +11, +14, +21, +22	unknown	1784	
0154	common B-ALL	m	78	initial	78	blood				+1, del(6)(q21q23), del(11)(q21q23), +19, +21, +21	dead	79	
0161	common B-ALL	m	23	initial	23	bone marrow				+6, +11, +14, +21, +21	unknown	1853	
0168	common B-ALL	f	78	initial	78	blood				−4, −7, −9, t(12;17)(q24;q21), −17, +mar	alive	1997	
0183	common B-ALL	f	59	initial	59	bone marrow					dead	1396	
0188	common B-ALL	f	80	relapse	78	blood				t(4;9)(p15;q21), −4, −7, −9, t(12;17)(q24;q12), −17, +mar	alive	1994	
0202	common B-ALL	m	36	initial	36	bone marrow					unknown	343	✓
0207	common B-ALL	m	46	initial	46	bone marrow					unknown	959	
0212	common B-ALL	f	62	relapse	62	bone marrow					alive	172	✓
0213	common B-ALL	m	53	initial	53	bone marrow					alive	44	
P67	common B-ALL	m	18	initial	18	bone marrow					alive	5180	✓
P74	common B-ALL	f	76	initial	76	unknown					unknown	287	
0141	common B-ALL, biphenotypic	m	43	initial	43	bone marrow				+21	alive	2880	✓
P33	pre-B-ALL	m	28	initial	28	unknown					unknown	274	✓
0094	mature B-ALL	f	85	relapse	84	blood					dead	226	✓
0074	B-ALL, not further specified	m	31	initial	31	blood				del(1)(q23), del(12)(p13), +21	dead	35	✓
0138	B-ALL, not further specified	f	49	initial	49	bone marrow				der(15), MYC rearrangement, TCF3 deletion	dead	46	
0151	B-ALL, not further specified	m	84	initial	84	blood					dead	97	✓
0159	B-ALL, not further specified	f	74	initial	74	bone marrow					unknown	-	✓
0200	B-ALL, not further specified	f	37	initial	37	blood					unknown	5	✓
0204	B-ALL, not further specified	m	20	initial	20	blood				CRLF2 and IGH rearrangement	alive	724	
P21	B-ALL, not further specified	f	76	relapse	75	blood					unknown	2	
P11	B-ALL, not further specified	f	22	relapse	19	bone marrow					dead	1384	
P25	pre T-ALL	m	22	initial	22	bone marrow				add(3)(q), del(7)(q31), +21	unknown	3561	
0019	T-ALL, not further specified	m	26	initial	26	bone marrow					unknown	2933	✓
0070	T-ALL, not further specified	m	43	initial	43	blood					unknown	2094	
0170	T-ALL, not further specified	f	57	initial	57	bone marrow					alive	1890	✓

m, male; f, female. Cytogenetics: red fields indicate a positive result of the analysis, while grey fields indicate absence of the respective aberration. Fields were left white when no analysis was conducted, or the result is unknown. Ph+, *BCR::ABL1*-positive; MLL, *KMT2A* rearrangement. Survival: days after diagnosis are calculated with either the date of death or the last day of contact. Unknown condition was stated for samples where the last contact was more than twelve months ago. Xeno: red fields indicate samples that were subsequently xenografted into NSG mice for further evaluation. Arrows indicate successful engraftment and crosses failed attempts of in vivo tumor cell proliferation.

**Table 2 cells-11-00150-t002:** All variants detected during the serial transplantation of *KMT2A*-positive primary samples.

			#0054		#0094					#0122				#0134				#0152			#0159			
Gene	Base Change	AA Change	Prim	P1	Prim	P1	P2	P3	P4	Prim	P1	P2	P3	Prim	P1	P2	P3	Prim	P1	P2	Prim	P1	P2	P3
APC	c.4326T > A	p.(=)	0	0	51	45	45	45	46	0	0	0		48	48	45		0	0	0	0	0	0	0
												0	0			45	45							
APC	c.4479G > a	p.(=)	0	0	100	100	100	100	100	49	52	50		100	99	100		100	100	100	0	0	0	0
												50	51			100	100							
ATM	c.*29C > G	p.?			0	0	0	0	0	0	0	0		0	0	0		0	0	0	49	50	50	53
			0	0								0	0			0	0							
ATM	c.5793T > C	p.(=)			0	0	0	0	0	0	0	0		0	0	0		0	0	0	52	50	50	51
			0	0								0	0			0	0							
BRAF	c.1789C > T	p.(=)			47	52	51	49	47	0	0	0		51	48	49		0	0	0	0	0	0	0
			0	0								0	0			46	50							
CSF1R	c.*36CA > TC	p.?	100	100	100	100	100	100	100	34	29	31		100	100	100		36	31	28	100	100	100	100
												13	30			100	100							
EGFR	c.2361G > A	p.(=)	51	51	100	100	100	100	100	53	44	52		100	100	100		100	100	100	51	49	49	48
												48	47			100	100							
ERBB4	c.421+58A > G	p.?	59	59	0	0	0	0	0	100	100	100		0	0	0		0	0	0	0	0	0	0
												100	100			0	0							
ERBB4	c.884-20T > C	p.?	2	0	0	0	0	0	0	0	0	0		0	0	0		0	0	0	0	0	0	0
												0	0			0	0							
FBXW7	c.1524A > G	p.(=)	0	0	0	0	0	0	0	3	0	0		0	0	0		0	0	0	0	0	0	0
												0	0			0	0							
FGFR3	c.1953G > A	p.(=)	100	100	0	0	0	0	0	100	100	100		100	100	100		0	0	0	0	0	0	0
												100	100			100	100							
FGFR3	c.1959+22G > A	p.?	0	0	0	0	0	0	0	0	0	0		0	0	0		0	0	0	0	0	0	0
												49	50			0	0							
FLT3	c.1310-3T > C	p.?	100	100	100	100	100	100	100	0	0	0		100	100	100		100	100	100	51	52	52	52
												0	0			100	100							
FLT3	c.1775T > C	p.Val592Ala	0	0	44	50	51	51	49	0	0	0		51	52	49		0	0	0	0	0	0	0
												0	0			56	52							
FLT3	c.2504A > T	p.As p835Val	0	0	0	0	0	0	0	3	0	0		0	0	0		0	0	0	0	0	0	0
												0	0			0	0							
HRAS	c.81T > C	p.(=)	0	0	60	49	54	48	48	52	49	52		48	53	51		55	55	46	0	0	0	0
												50	51			50	50							
IDH1	c.315C > T	p.(=)	0	0	0	0	0	0	0	0	0	0		0	0	0		0	0	0	51	53	53	52
												0	0			0	0							
JAK3	c.2164G > A	p.Val722le	0	0	0	0	0	0	0	0	0	0		0	0	0		0	0	0	51	49	49	52
												0	0			0	0							
KDR	c.798+54G > A	p.?	0	0	52	44	48	45	47	49	50	45		46	48	52		100	100	100	100	98	98	100
												48	46			50	51							
KDR	c.4008C > T	p.(=)	0	0	0	0	0	0	0	49	53	48		0	0	0		0	0	0	0	0	0	0
												49	48			0	0							
KIT	c.1638A > G	p.(=)	51	50	0	0	0	0	0	0	0	0		0	0	0		0	0	0	0	0	0	0
												0	0			0	0							
KRAS	c.34G > A	p.Gly 12Ser	0	0	0	0	0	0	0	31	24	4		0	0	0		0	0	0	0	0	0	0
												6	7			0	0							
MET	c.3029C > T	p.Thr 1010lle	0	0	0	0	0	0	0	0	0	0		0	0	0		0	0	0	51	50	50	50
												0	0			0	0							
NRAS	c.183A > C	p.Gln61His	0	0	0	0	0	0	0	0	0	0		0	0	0		0	0	0	6	0	0	0
												0	0			0	0							
PDGFRA	c.1701A > G	p.(=)	100	100	100	100	100	100	100	100	100	100		100	100	100		100	100	100	100	100	100	100
												100	100			100	100							
PDGFRA	c.2472C > T	p.(=)	49	47	0	0	0	0	0	0	0	0		0	0	0		0	0	0	0	0	0	0
												0	0			0	0							
PIK3CA	c.352+40A > G	p.?	0	0	0	0	0	0	0	0	0	0		0	0	0		0	0	0	51	53	53	51
												0	0			0	0							
PIK3CA	c.1636C > A	p.Gln546Lys	0	0	0	0	0	0	0	0	0	0		0	0	0		0	0	0	4	0	0	0
												0	0			0	0							
PIK3CA	c.3075C > T	p.(=)	0	0	0	0	0	0	0	0	0	0		0	0	0		0	0	0	51	50	50	52
												0	0			0	0							
RET	c.2307G > T	p.(=)	48	51	54	49	52	54	50	0	0	0		51	50	54		54	50	49	100	100	100	100
												0	0			51	52							
RET	c.2712C > G	p.(=)	0	0	0	0	0	0	0	0	0	0		0	0	0		51	50	50	51	47	47	49
												0	0			0	0							
SMAD4	c.955+58C > T	p.?	54	49	48	51	54	54	56	0	0	0		53	53	48		0	0	0	0	0	0	0
												0	0			51	48							
STK11	c.126G > C	p.(=)	0	0	0	0	0	0	0	0	0	0		5	0	0		0	0	0	0	0	0	0
												0	0			0	0							
STK11	c.465-51T > C	p.?	100	100	0	0	0	0	0	0	0	0		0	0	0		0	0	0	100	100	100	100
												0	0			0	0							
TP53	c.215C > G	p.Pro72Arg	99	100	94	49	49	50	50	100	100	100		52	53	53		100	100	99	56	52	52	49
												98	99			51	61							
TP53	c.847C > A	p.Arg283Ser	0	0	86	0	0	0	0	0	0	0		0	0	0		0	0	0	0	0	0	0
												0	0			0	0							

All variants detected are summarized irrespectively of their pathogenicity. Numbers indicate the allele frequency (%). AA, amino acid; p.(=), no change in protein translation; p.?, extraexonic variant; Prim, primary tumor; P, passage. The color of the boxes indicates the allele frequency, with low frequencies displayed in red and high frequencies in blue shade. For patients #0122 and #0134, two individual mice were analyzed in passage 2. The allele frequencies of these animals are listed on top of each other. The cells of the second (lower) mouse were used for xenotransplantation and the generation of passage 3.

## Data Availability

The datasets used and/or analyzed during the study are available from the corresponding author upon reasonable request.

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
