# Peer review of "The Molecular Subtype of Adult Acute Lymphoblastic Leukemia Samples Determines the Engraftment Site and Proliferation Kinetics in Patient-Derived Xenograft Models"

_cells, 2022, doi:10.3390/cells11010150_

Round 1

Reviewer 1 Report

Richter at al. describe the establishment of a leukemia ALL, annotating egraftment rates in the different molecular subtypes.

They report a higher engraftment rate of models harboring BCR-ABL1 or MLL translocations and correlation with with poor prognosis of patients. The collect data over the bona fide developed models, which show overall a stability as assessed by STR profiling, across different generations.

The manuscript is very well written and structured to the point. Data are clearly presented in an effective manner.

Can the authors comment on the standards currently available in the field of PDX models? What is an acceptable genetic variation across generation? In a recent Nature Genetics paper, PDX were shown to be not significantly affected from the passage human-mouse host. Can the author comment relative to this?

Have they tested complets CNA profile of their biobank across generations? Are ALL PDX models different than solid tumors in terms of genetic instability?

Author Response

We thank reviewer 1 for the careful revision of our manuscript as well as the suggestions for improvement. We have now revised the paper and think that it benefits from the recommendations made by reviewer 1, namely the more detailed discussion of a suggested paper and genetic stability of our model systems. All changes are documented in track changes mode in the revised version of the paper. We hope that the manuscript is now suitable for publication in Cells.

Reviewer 1: “Can the authors comment on the standards currently available in the field of PDX models? What is an acceptable genetic variation across generation? In a recent Nature Genetics paper, PDX were shown to be not significantly affected from the passage human-mouse host. Can the author comment relative to this?”

Answer: We are aware of the mentioned Nature Genetics paper by Woo et al. (doi: 10.1038/s41588-020-00750-6) and have already compared our results to the findings of this study (please see lines 511-513). It is to be noted, however, that the samples in the mentioned cohort were analyzed for copy number alterations (CNA) while our PDX and primary samples underwent targeted Cancer Hotspot Panel gene sequencing. As those methods cannot be compared directly because they focus on different mechanisms of variance evaluation, it is indeed difficult to state if our findings are comparable to those of the Woo study. The more important difference between the Woo paper and our manuscript is that only solid tumor samples were investigated in the Nature Genetics paper while we focused on ALL specimens. Due to higher intratumoral heterogeneity and non-orthotopic engraftment in the solid models, different mechanisms related to microenvironment, circulation and clonality may interfere with a direct comparison of the two studies, and make it even more difficult to list standards of genetic stability. Few studies have evaluated the genetic stability of PDX models in ALL cohorts, and our results have been compared to those in the discussion section (lines 527-529). We believe that this approach is more relevant than a comparison to solid cancer samples. To put more emphasize on the technical differences of the studies, we inserted a limiting sentence in the discussion section, lines 513-516.

Reviewer 1: “Have they tested complets CNA profile of their biobank across generations?”

Answer: Unfortunately, we have not yet performed CNA profiling for all but one primary tumor sample. This is without doubt a very interesting thing to do in the future, especially due to the lack of any studies investigating CNA in ALL PDX models.

Reviewer 1: “Are ALL PDX models different than solid tumors in terms of genetic instability?”

Answer: This question is very difficult to answer. A comprehensive approach analyzing primary and PDX samples of solid and liquid tumor specimens derived from the same lab and using the same sequencing and bioinformatics workflow would be necessary to exclude most technical sources of possible biases. We are not aware of any comparative study investigating genetic stability of PDX models in the described setting for adults. In the pediatric setting, however, Rokita et al. (10.1016/j.celrep.2019.09.071) investigated PDX models of solid and hematological malignancies, but not over several passages. They showed faithful disease recapitulation across all PDX models.

Reviewer 2 Report

In this study, Richter and colleagues describe the generation of a broad panel of patient-derived xenografts (PDX) established using samples obtained from adult patients with acute lymphoblastic leukemia (ALL). A sample bank of 44 patients, (37 Dx, 7 relapse patients), covering a range of molecular subtypes (including 7 MLL and 14 BCR-ABL1) was used as the source of primary material. Of the 44 patients, 18 were selected for engraftment in PDX (in 31 total mice). The bulk of the report describes the characteristics and stability of the initial and serially transplanted PDX ALL populations, which are analyzed in reasonable depth.

The primary contribution of this study is the description of a highly successful (94%) engraftment protocol for adult ALL. This success rate is in contrast to the few previous adult ALL PDX reports, and notably was achieved in the absence of any preconditioning of the mice. Again, this contrasts earlier reports and indicates that effective PDX generation for adult ALL is possible, which would enable more experimental interrogation than is currently possible with the limited models available. However, all the methods used, while appropriate, have been applied in previously PDX studies and no novel biological insights arising from the PDX models are provided by the study.

The manuscript is very well-written (the quality of the methods description is especially noteworthy) and puts the work in an appropriate context with good clarity. It represents a meaningful contribution to the field.

Primary comments:

  1. It is stated that the engraftment data “is only representative for the given sample collection and remains to be validated as only two T-ALL patient samples and three samples with normal karyotypes were included in the study cohort”. But subtype analysis remains the central component of the study, despite the lack of numbers. A meaningful comparison of MLL vs BCR-ABL is feasible with this sample bank, but other comparisons should be limited. The continued comparison of subtypes raises the risk of over-interpretation. 
  2. The specific xenografts are unclear. It would be very helpful to have a table (or an amended Table 1) that shows clearly the patient samples, subtypes, number of mice engrafted for each, and the number of ALL cells engrafted. As mice receiving the same sample will mostly show similar outgrowth, this is important to allow a clear interpretation of the results. Coloured dots are shown in Fig. 2 for this purpose – this is helpful (but quite challenging) - and does appear to show the over-representation of a small number of patient samples. The authors should clarify this data presentation. In this regard, it is a weakness that the maximal number of MLL and BCR-ABL samples were not engrafted to allow a well-powered comparison between those two subtypes.
  3. Clonal selection is strongly implicated in the ALL outgrowth in PDX, which has significant implications for the relevance of the expanded population for clinical progression (and hence for pre-clinical investigation). The limitations of the targeted sequencing and very limited flow phenotyping should be discussed in this context, as neither has the resolution to demonstrate that the outgrowing populations in mice are relevant to the patient. Comparison with later clinical samples would allow this to be addressed, but is likely not feasible at this point.
  4. The inclusion of an early timepoint would have been very informative for the interpretation of engraftment kinetics. By waiting until a relatively high burden was reached, details of differences in initiating cell number and leukemia-initiating capacity will be lost behind the different inherent proliferation rates of dominant expanding populations. The rationale for choosing a single endpoint, and the limitations of that approach, should be discussed.

Minor Points:

  1. Title: change Ki-Netics to Kinetics
  2. Are mouse cell counts accurate in presence of high human cell burden? Flow analysis to confirm the expansion of granulocytes would be helpful.
  3. The mice were older than is generally used for PDX engraftment. Do the authors have a reason why they chose this age range?
  4. For the only non-engraving sample, was the prince of ALL blasts in spleen or bone marrow investigated at endpoint.

Author Response

We thank Reviewer 2 for the careful revision of our manuscript as well as the suggestions for improvement. We are pleased to see that a key part of the manuscript, namely a detailed description of the methodology which allowed us to achieve improved engraftment rates, was noted by the Reviewer. We have now revised the paper accordingly and think that it greatly benefits from the recommendations made by Reviewer 2. All changes are documented in track changes mode in the revised version of the paper. We hope that the manuscript is now suitable for publication in Cells.

Reviewer 2: “It is stated that the engraftment data “is only representative for the given sample collection and remains to be validated as only two T-ALL patient samples and three samples with normal karyotypes were included in the study cohort”. But subtype analysis remains the central component of the study, despite the lack of numbers. A meaningful comparison of MLL vs BCR-ABL is feasible with this sample bank, but other comparisons should be limited. The continued comparison of subtypes raises the risk of over-interpretation.”

Answer: We strongly agree with Reviewer 2 in this point and limited our interpretation at several spots in the manuscript, including the abstract (line 27), results section (lines 301-304 and lines 328-330) and discussion part (lines 493-495).

Reviewer 2: “The specific xenografts are unclear. It would be very helpful to have a table (or an amended Table 1) that shows clearly the patient samples, subtypes, number of mice engrafted for each, and the number of ALL cells engrafted. As mice receiving the same sample will mostly show similar outgrowth, this is important to allow a clear interpretation of the results. Coloured dots are shown in Fig. 2 for this purpose – this is helpful (but quite challenging) - and does appear to show the over-representation of a small number of patient samples. The authors should clarify this data presentation. In this regard, it is a weakness that the maximal number of MLL and BCR-ABL samples were not engrafted to allow a well-powered comparison between those two subtypes.”

Answer: We have summarized the desired data in Table S1, entitled “Total numbers of animals per passage for each primary sample and respectively injected cell counts.” and referred to this table in the methods section (lines 135-136). From this table, it can be drawn that for both, KMT2A and BCR::ABL1 positive samples, a total of 6-9 animals were injected per primary sample. For KMT2A cases, five out of six samples were injected in 8 or 9 animals and one was injected in 6. For BCR::ABL1 cases, three out of five were injected in 6 animals, one cases was injected in 7 animals and one case in 8 animals. For the T-ALL samples, cells were injected in 5 and 8 animals. The variance of injected animals within the respective subtypes and also between subgroups is therefore rather small, in our opinion allowing for the demonstrated comparison between those entities. The observation of Reviewer 2, stating that few samples are over-represented, is most likely caused by the fact that some colors in Figure 2 appear very similar and can easily be mixed up. Unfortunately, we do not see a satisfying alternative to present those data. We hope that the addition of Table S1 is satisfactory for this problem.

Reviewer 2: “Clonal selection is strongly implicated in the ALL outgrowth in PDX, which has significant implications for the relevance of the expanded population for clinical progression (and hence for pre-clinical investigation). The limitations of the targeted sequencing and very limited flow phenotyping should be discussed in this context, as neither has the resolution to demonstrate that the outgrowing populations in mice are relevant to the patient. Comparison with later clinical samples would allow this to be addressed, but is likely not feasible at this point.”

Answer: We agree with Reviewer 2 that clonal selection does occur during PDX establishment. Some studies in AML and myeloma, however, have demonstrated that the clones arising in PDX models are similar to those of relapsed patients, suggesting that xenografting induces a comparable clonal pressure on the tumor cells compared to cancer treatment. We now discuss this interesting finding in the discussion section, lines 519-526. The limitations of the performed genotyping are further discussed in lines 513-516. The suggested comparison with later clinical samples indeed presents a very interesting experiment. Unfortunately it is, as Reviewer 2 correctly noted, behind the scope and feasibility of the present manuscript.

Reviewer 2: “The inclusion of an early timepoint would have been very informative for the interpretation of engraftment kinetics. By waiting until a relatively high burden was reached, details of differences in initiating cell number and leukemia-initiating capacity will be lost behind the different inherent proliferation rates of dominant expanding populations. The rationale for choosing a single endpoint, and the limitations of that approach, should be discussed.”

Answer: Similar to the suggestion to include later clinical specimens, this approach would as well be very interesting and result in increased knowledge about engraftment kinetics. In the present work, our focus was mainly on the engraftment sites and genetic alterations. An earlier experiment termination would result in reduced blast frequencies in bone marrow, spleen and blood, thus increasing the murine background in molecular analyses and complicating data analysis. Further, fewer backup samples of bone marrow and spleen blasts could be collected and those would be of reduced purity, introducing a potential technical bias for the evaluation of proliferation kinetics throughout serial passaging. Still, we agree that an earlier time point could have resulted in the detection of further subclones, albeit probably not with the sequencing panel used in our study. We added respective sentences in the discussion section, lines 513-518.

Reviewer 2: “Title: change Ki-Netics to Kinetics”

Answer: We apologize for this obvious error and have changed the title.

Reviewer 2: “Are mouse cell counts accurate in presence of high human cell burden? Flow analysis to confirm the expansion of granulocytes would be helpful.”

Answer: For the analysis of murine cell counts we used an ADVIA system with respective controls specifically for the analysis of murine blood samples. It is behind our knowledge, however, if high human cell counts interfere with this analysis. Flow cytometry to confirm the results would have indeed been a good thing to do. On the other hand, as the total number of granulocytes increased, we do not think that this is a cross-reaction with the human cells, as those are obviously lymphocytes.

Reviewer 2: “The mice were older than is generally used for PDX engraftment. Do the authors have a reason why they chose this age range?”

Answer: We routinely use 8-10 week old mice for xenografting and in some cases older animals. This is due to the fact that in other, cell line-based xenografting experiments, we found that mice of this age are grown out and tolerate the procedure including bioluminescence imaging with repeated narcotization without problems. Younger animals with lower body weight sometimes had problems during anesthesia. We therefore usually order animals in the given age span, also to keep experiments comparable and reduce changes in experimental conditions. However, sometimes planned injections need to be rescheduled due to different reasons, resulting in the mice growing a little bit older than anticipated. In those cases we still use the animals because it would be unethical not to.

Reviewer 2: “For the only non-engraving sample, was the prince of ALL blasts in spleen or bone marrow investigated at endpoint.”

Answer: For sample #0031, two mice were injected and observed for 191 days. As no signs of tumor cell proliferation were detected by flow cytometry in the blood, mice were euthanized. Unfortunately, we did not determine the human tumor cell frequency in bone marrow and blood. Spleen measurement, however, was performed and spleens resembled the phenotype of healthy, non-leukemic NSG mice with a length of 1.4 cm, width of 0.3 cm and weight of 0.05 g.